# Intravenous Oncolytic Vaccinia Virus Therapy Results in a Differential Immune Response between Cancer Patients

**DOI:** 10.3390/cancers14092181

**Published:** 2022-04-27

**Authors:** Emma J. West, Karen J. Scott, Emma Tidswell, Kaidre Bendjama, Nicolas Stojkowitz, Monika Lusky, Marta Kurzawa, Raj Prasad, Giles Toogood, Christy Ralph, D. Alan Anthoney, Alan A. Melcher, Fiona J. Collinson, Adel Samson

**Affiliations:** 1Leeds Institute of Medical Research at St. James’s, University of Leeds, Leeds LS9 7TF, UK; e.j.west@leeds.ac.uk (E.J.W.); k.j.scott@leeds.ac.uk (K.J.S.); e.l.tidswell@leeds.ac.uk (E.T.); 2Transgene, 67082 Strasbourg, France; bendjama@transgene.fr (K.B.); stojkowitz@transgene.fr (N.S.); monikaluskyhelm@gmail.com (M.L.); 3Leeds Teaching Hospitals NHS Trust, Leeds LS9 7TF, UK; marta.kurzawa@nhs.net (M.K.); raj.prasad@nhs.net (R.P.); giles.toogood@nhs.net (G.T.); christy.ralph@nhs.net (C.R.); alan.anthoney@nhs.net (D.A.A.); f.j.collinson@leeds.ac.uk (F.J.C.); 4The Institute of Cancer Research, London SW7 3RP, UK; alan.melcher@icr.ac.uk

**Keywords:** vaccinia virus, oncolytic virus therapy, immunotherapy, biomarkers, interferon response, immune checkpoint blockade, immune exhaustion, differential immune response

## Abstract

**Simple Summary:**

Oncolytic viruses (OVs) have been extensively studied as an immunotherapeutic agent against a variety of cancers with some successes. Immunotherapeutic strategies, such as OVs, aim to transform an immunologically ‘cold’ tumour microenvironment into a more favourable inflammatory ‘hot’ tumour. However, it is evident that not all patients have a favourable response to treatment. Furthermore, reliable biomarkers able to predict a patient’s response to therapy have not yet been elucidated. We show evidence of a distinct immunologically exhausted profile in patients who do not respond to OV, which may pave the way for the development of predictive biomarkers leading to a more personalised approach to cancer treatment using combination therapies.

**Abstract:**

*Pexa-Vec* is an engineered Wyeth-strain vaccinia oncolytic virus (OV), which has been tested extensively in clinical trials, demonstrating enhanced cytotoxic T cell infiltration into tumours following treatment. Favourable immune consequences to *Pexa-Vec* include the induction of an interferon (IFN) response, followed by inflammatory cytokine/chemokine secretion. This promotes tumour immune infiltration, innate and adaptive immune cell activation and T cell priming, culminating in targeted tumour cell killing, i.e., an immunologically ‘cold’ tumour microenvironment is transformed into a ‘hot’ tumour. However, as with all immunotherapies, not all patients respond in a uniformly favourable manner. Our study herein, shows a differential immune response by patients to intravenous *Pexa-Vec* therapy, whereby some patients responded to the virus in a typical and expected manner, demonstrating a significant IFN induction and subsequent peripheral immune activation. However, other patients experienced a markedly subdued immune response and appeared to exhibit an exhausted phenotype at baseline, characterised by higher baseline immune checkpoint expression and regulatory T cell (Treg) levels. This differential baseline immunological profile accurately predicted the subsequent response *to Pexa-Vec* and may, therefore, enable the development of predictive biomarkers for *Pexa-Vec* and OV therapies more widely. If confirmed in larger clinical trials, these immunological biomarkers may enable a personalised approach, whereby patients with an exhausted baseline immune profile are treated with immune checkpoint blockade, with the aim of reversing immune exhaustion, prior to or alongside OV therapy.

## 1. Introduction

Oncolytic viruses (OVs) are immunotherapeutic agents that preferentially replicate in and kill malignant cells. These OVs are non-pathogenic and, dependent on the viral species, can be engineered to express immune-stimulating or targeted transgenes [1]. A number of OVs have been extensively tested in clinical trials, three of which are currently licensed for routine treatment in cancer patients [2].

OVs exert a multitude of effects on the immune system, including the release of inflammatory cytokines, thereby enhancing innate anti-cancer function. Principal amongst these are interferons (IFNs) [3,4], which induce the expression of hundreds of IFN-stimulated genes [5]. The secreted inflammatory cytokines and chemokines promote innate immune cell activation, both within the tumour and systemically, as well as immune cell infiltration into the tumour. Targeted infection of tumour cells results in immunogenic tumour cell death, phagocytosis of tumour-associated antigens (TAAs) by antigen-presenting cells (APCs), and enhanced anti-cancer T cell priming [6]. The overall result is that an immunologically ‘cold’ tumour is converted into a ‘hot’ tumour.

An unwelcome effect of the inflammatory environment induced by OVs is increased expression of immune checkpoint molecules, e.g., programmed cell death protein (PD)-1 and PD-ligand (PD-L)-1 [7,8], which act to dampen immune activation and T cell priming. This evolved mechanism to hinder autoimmunity may also impede the anti-cancer efficacy of OVs. Nonetheless, with the advent of clinically available immune checkpoint blockade (ICB), this immunosuppressive mechanism can be overcome by combining ICB with OV therapy. Multiple pre-clinical and early-phase clinical trials combining OVs with anti-PD-1/anti-PD-L1 have produced early signs of clinical benefit [9,10].

OV therapy has been linked with elevated expression of other immune checkpoint molecules, including cytotoxic T lymphocyte-associated protein 4 (CTLA-4). Clinical trials in patients with advanced solid malignancies have indicated the superiority of combination OV and anti-CTLA-4 approach over monotherapy; the anti-CTLA-4 antibody, ipilimumab, has been used in conjunction with many OVs including Talimogene Laherparepvec [11,12] and coxsackievirus A21 [13] with encouraging results. A preclinical study combining Newcastle Disease virus with anti-CTLA-4 showed evidence of enhanced responses over monotherapy [14]. 

CTLA-4 is constitutively expressed at very high levels on regulatory T cells (Tregs); elevated levels being essential for the functionality of Tregs [15]. Tregs are potent suppressors of effector T cell function via multiple mechanisms, for example, limiting interleukin (IL)-2 bioavailability, CTLA-4-mediated downregulation of costimulatory molecules on APCs, plus secretion of immunoinhibitory cytokines, such as IL-10 or transforming growth factor-β (TGF-β) [16,17]. Tregs have a high prevalence in the context of cancer; in fact, the density of Tregs within the tumour microenvironment (TME) is predictive of poor clinical outcome, suggesting that Tregs may play a functional role in cancer progression. In addition, a higher frequency of peripheral Tregs has also been linked to adverse survival [18]. The use of blocking antibodies against CTLA-4 can reduce the suppression mediated by tumour-infiltrating Tregs, thereby restoring proliferation and cytokine production by effector T cells [19]. 

The majority of OV studies thus far have focused on safety and determining maximum tolerated doses, whilst very few have sought to determine biomarkers predictive of therapeutic success. However, one such study investigating the therapeutic response to vaccinia revealed that a significant reduction in immunoglobulin-like transcript 2(ILT2)-expressing Tregs after treatment was associated with response to therapy [20], whereas Liikanen et al. [21] suggested that a low level of circulating high mobility group box 1 (HMGB1) could predict response to OVs. Furthermore, common immunological themes have emerged between patients who appear to respond to a greater extent to OVs than others. Taipale et al. [22] observed that patients with worse outcomes following adenovirus treatment appeared to exhibit a higher degree of pre-existing immune response, such as higher proliferation and quantity of lymphocytes at baseline and gene signatures involved in, e.g., IFN signaling, B cell receptor signaling, and innate immunity. More recently, active trials are currently recruiting patients with defective mismatch repair (dMMR) tumours, as it is thought that a higher tumour mutational burden will lead to a higher tumour immunogenicity and optimum anti-cancer effects (NCT03767348).

Biomarkers for other immunotherapeutic strategies are more advanced, for example, patients who responded to combination T-VEC and anti-PD-1 had elevated CD8+ T cells, PD-L1, and IFNγ gene expression [10]; however, response did not appear to be associated with baseline tumour CD8+ T cells or IFNγ signature. In contrast, Tumeh et al. correlated a higher CD8 T cell density at the tumour margins with response to pembrolizumab [23]. Despite these advances in tumour-associated biomarkers, peripheral blood biomarkers would provide a non-invasive, easily obtainable indication to predict response to treatment. Some initial studies into blood markers indicate that elevated peripheral Tregs are associated with response to ipilimumab [24], whereas high major histocompatibility complex (MHC) class II gene expression was associated with clinical response to anti-PD-1 or –PD-L1 therapy [25]. 

*Pexa-Vec* (*Pexastimogene Devacirepvec*; JX-594) is an engineered Wyeth-strain vaccinia virus [26], which has been extensively studied in clinical trials. *Pexa-Vec* is engineered to express human granulocyte-macrophage colony stimulating factor (hGM-CSF), which positively impacts the immune system by, for example, stimulation, recruitment, and development of dendritic cells (DCs) [27]. The anti-cancer mechanisms of action of *Pexa-Vec*, including enhancement of cytotoxic T lymphocyte infiltration into tumour and anti-tumour immunity have been described in both in vitro and clinical studies [26,28,29,30,31], with hundreds of patients with advanced cancer having been treated. *Pexa-Vec* is currently being used in combination with ICB in many cancer types, for example, anti-PD-1 (NCT03071094, NCT03294083), anti-CTLA-4 (NCT02977156), and anti-PD-L1 (NCT03206073) and we await the results of these trials.

*Pexa-Vec* therapy is associated with an inflammatory response [30] and T cell infiltration in tumours [26]. The profile of cytokines/chemokines following *Pexa-Vec* therapy might indicate which patients are likely to respond more favourably, as other immunotherapy studies have hypothesised that quantitative changes in cytokine levels during treatment may be associated with survival [32]. Current prognosis for patients with colorectal cancer with liver metastases (CRCLM) is 50% survival at five years following liver resection [33], whereas 60% of patients with metastatic melanoma to the lymph nodes will relapse following surgical resection and adjuvant treatment [34].

Herein, we detail the characteristics of a differential peripheral blood and tumour immune response, following a single intravenous (i.v.) infusion of *Pexa-Vec* in nine cancer patients ahead of planned surgical resection of metastatic colorectal cancer or melanoma. Despite identical treatment for all nine patients, a stark differential immune response to the virus became apparent. In four patients, a significant and classical IFN-driven response was observed, as would be expected following i.v. oncolytic virus therapy [3,4], in contrast to a very minor or absent response from the remaining five patients, who appeared to have an exhausted immune profile at baseline. Whilst our patient cohort is small in size and the lack of paired tumour biopsies prevented correlation to the tumour microenvironment, these observations are worthy of further investigation in larger clinical trials using *Pexa Vec* and other OVs, to determine whether patients with a peripheral blood immune profile observed in the patient group who responded favourably to OV therapy are more likely to experience favorable clinical outcomes. If confirmed, these observations will greatly improve the treatment strategies for these patients and potentially pave the way to personalise and optimise the immunotherapeutic anti-cancer effects of *Pexa Vec* and other OVs, based on baseline biomarkers predictive of the anti-tumour immune response.

## 2. Materials and Methods

### 2.1. Experimental Design

EudraCT number 2012-000704-15. This was an open-label, non-randomised study of *Pexa-Vec* given as a one-hour i.v. infusion to patients prior to a planned surgical resection of tumour. Six patients with CRCLM and three metastatic melanoma patients were recruited and treated, after written, informed consent was obtained. Patients received a single dose of *Pexa-Vec* at 1 × 10^9^ (plaque-forming units) pfu 14 days (±4 days) prior to surgery. Eight patients had their planned surgery; one exhausted patient had their surgery cancelled when an up-to-date CT scan revealed pulmonary metastases. 

### 2.2. Pexa-Vec

*Pexa-Vec* (Pexastimogene Devacirepvec; JX-594) is a replication-competent, transgene-armed therapeutic vaccinia virus provided by Transgene S.A, France. *Pexa-Vec* is engineered for viral thymidine kinase gene inactivation and expression of hGM-CSF and β-galactosidase transgenes under the control of the synthetic early-late and p7.5 promoters, respectively. *Pexa-Vec* was stored at 1 × 10^9^ pfu/mL at −80 °C for use in in vitro experiments.

### 2.3. Patient Samples

Blood and tissue samples were collected, processed, and analysed using the Translational Cancer Immunotherapy Team quality-assured lab manual, which included standard operating procedures to regulate all processes.

Peripheral blood was collected into K_3_EDTA vacuette tubes (Greiner, Kremsmünster, Austria) and processed within 2 h of venepuncture. Blood samples were taken on day 1 (pre-infusion (D1 pre) and one-hour post-infusion), day 2 (D2), on the day of surgery, 1 month post-surgery and 3 months post-surgery. Tumour was obtained from planned surgical resections. 

### 2.4. Isolation of Peripheral Blood Mononuclear Cells (PBMCs), Plasma, and Serum from Peripheral Blood

Plasma was obtained from whole blood collected in K_3_EDTA vacutainers by centrifugation for 10 min at 2000× *g*. Aliquots were stored at −80 °C. 

K_3_EDTA blood was used to isolate PBMCs by density-gradient separation over lymphoprep™ (Axis Shield, Dundee, UK) as per manufacturer’s instructions. Cells were frozen at 1 × 10^7^/mL in 40% *(v*/*v*) Roswell Park Memorial Institute medium (RPMI)) containing 5 mM L-Glutamine and 1 mM sodium pyruvate (all Sigma, Dorset, UK), 50% (*v*/*v*) pooled human serum (HS; BioIVT, West Sussex, UK) and 10% (*v*/*v*) dimethyl sulphoxide (DMSO; Sigma). PBMCs were stored in liquid nitrogen.

### 2.5. Full Blood Counts

Full blood counts (FBCs) were performed as part of standard clinical care, where appropriate, at St. James’s University Hospital. The Patient Pathway Manager and Results Server systems were used to obtain total lymphocyte counts (expressed as 10^9^/L) throughout treatment. Normal ranges of lymphocytes were defined by St James’s University Hospital as 1–4.5 × 10^9^/L.

### 2.6. Immunohistochemistry (IHC)

Formalin-fixed paraffin-embedded tissue obtained from surgical resection of patient tumours was used for IHC analysis. Tumours were processed using an automated Bond Max system (Leica Biosystems, Milton Keynes, UK) as described [35]. Mouse-anti-human CD8 antibody (Dako) was used at 1:100 dilution, followed by anti-mouse secondary (Abcam, Cambridge, UK) at 1:500; CD8 positivity was detected using ImmPACT Vector Red (Vector Labs, Oxfordshire, UK). Control sections were processed without the addition of primary antibody. Digital images were acquired at ×20 magnification and quantified using ImageScope software (version 12.4.3.5008, Leica Biosystems).

### 2.7. Luminex

Bio-Plex Pro^TM^ Cytokine and Chemokine Assays (21-plex; human group I and 27-plex; human group II or 48-plex; human cytokine; all BioRad, Hertfordshire, UK) were used to detect levels of plasma cytokines/chemokines throughout treatment, as per manufacturer’s instructions. IFN-β was measured using the VeriKine-HS Human Interferon Beta ELISA Kit for plasma (R&D Systems, Abingdon, UK), as per manufacturer’s instructions. Data is expressed as absolute plasma concentration or relative fold change in post-treatment samples compared to pre-treatment samples. Statistical significance between Responder (black; *n* = 4) and Exhausted (white; *n* = 4/5) patients at specific time points was determined using Anova; *** *q*-value < 0.001, **** *q*-value < 0.0001. Paired T tests were used to compare D2 to baseline samples within each patient group (* *p* < 0.05, ** *p* < 0.01, *** *p* < 0.001).

### 2.8. NK Cell CD107 Degranulation Assay of Patient PBMCs

NK cell activation was assessed using a CD107 degranulation assay [36]. PBMCs from pre-treatment and post-*Pexa-Vec* infusion were co-cultured at a ratio of 1:1 with tumour-associated cell lines (Mel888 or SW620 for melanoma and CRCLM patients, respectively) for 1 h. Brefeldin A (1 µL/mL; Sigma) was added, and the co-culture continued for a further 4 h before PBMCs were stained for CD3-PerCP (SK7; BD Biosciences, Wokingham, UK), CD56-PE (AF12-7H3; Miltenyi, Bergisch Gladbach, Germany) and CD107a/b-FITC (H4A3; BD Biosciences). A CytoFLEX S flow cytometer was used to detect CD107 positivity; analysis was performed using CytExpert software (both Beckman Coulter, Buckinghamshire, UK). Data is expressed as fold-change difference from pre-treatment samples. Statistical significance is determined by unpaired T tests between Responder (black; *n* = 4) and Exhausted (white; *n* = 4/5) patients at specific time points (* *p* < 0.05).

### 2.9. Immunophenotyping of Patient PBMCs

PBMCs were stained for a panel of immune cell populations and specific activation markers prior to data acquisition on a CytoFLEX S and analysed using CytExpert software. Briefly, PBMCs were stained for CD3 (UCHT1; Pacific Blue), CD4 (13B8.2; Krome Orange), CD8 (B9.11; FITC), CD56 (N901; PC7), CD25 (B1.49.9; PC7), CD127 (SSDCLY107D2; APC-AF750), FoxP3 (259D; APC), γδTCR (IMMU510; FITC), CD19 (J3-119; Pacific Blue), CD14 (RM052; FITC), CD69 (TP.55.3; APC) and PD-L1 (APC-AF700) using a custom-designed panel of DURAClone tubes, in conjunction with appropriate isotype controls (all Beckman Coulter). Immune cell subsets were defined as: CD3 + CD4 + (CD4 T cells); CD3 + CD8 + (CD8 T cells); CD3-CD56 + (NK cells); CD3 + CD56 + (NKT cells), and CD14 + (monocytes). Immune cell frequency was calculated as % of cell population within total PBMCs. Positive expression of CD69 and PD-L1 was used to calculate fold-change differences in expression from pre-treatment samples. Statistical significance between Responder (black; *n* = 4) and Exhausted (white; *n* = 4/5) patients was determined using unpaired T tests, whereas paired T tests were used to compare D2 to baseline samples within each patient group (* *p* < 0.05, ** *p* < 0.01, *** *p* < 0.001).

### 2.10. Data Interpretation

Data are either: (a) presented as absolute values or (b) presented as fold-change in comparison to baseline to show the differential response of the two patient groups in response to treatment, a common strategy used previously with similar data sets [10].

## 3. Results

### 3.1. Patient Demographics

Nine patients were recruited with each patient receiving a single, 1 h i.v. infusion of 1 × 10^9^ pfu *Pexa-Vec*, 14 ± 4 days ahead of planned surgery to remove metastatic lesions (Table 1). Surgery was performed on all patients except E3, where surgery was cancelled following a CT scan showing disease progression. As a guide to prognosis for both Responder and Exhausted patient groups we have included the sum of the longest diameters of the tumours for each patient. This indicates that the tumour volumes were approximately similar between the two groups.

### 3.2. Differential Cytokine Secretion following Pexa-Vec Infusion

Peripheral blood samples were collected at baseline (D1 pre) and at specific time points following infusion (Figure 1A). Despite all nine patients receiving the same dose of virus, a differential immune response was evident. In four patients, a significant peak in cytokine production following virus infusion was observed; this peak was absent in the remaining five patients (Figure 1B,C, Appendix A). Collectively with the data in subsequent figures showing a similar phenomenon, we therefore labelled these two contrasting groups as ‘Responder’ and ‘Exhausted’. Overall, peripheral immune responses to *Pexa-Vec* peaked at D2, 24 h after virus infusion, for both the Responder and Exhausted groups. Specifically, a significant type I IFN (IFN-α and IFN-β) response to virus was apparent in Responders, whilst little or no increase in secreted IFN-β was observed in Exhausted patients at D2 (Figure 1B, Appendix A). Accordingly, a similar differential response was observed in the induction of IFN-stimulated inflammatory cytokines at D2: the T cell stimulants IL-2 receptor alpha chain (IL-2Rα) and IL-12p40 [37,38], the NK cell-stimulating cytokine IL-18 [39] and the pro-apoptotic cytokine TNF-related apoptosis-inducing ligand (TRAIL) [40], all increased significantly higher from baseline pre-*Pexa-Vec* concentrations in the Responder group, in comparison to the Exhausted group, although the absolute levels were lower or similar in comparison to the Exhausted group (Figure 1C, Appendix A).

### 3.3. Differential CD8 T Cell Tumour Infiltration following Pexa-Vec Infusion

In addition to a disparity in inflammatory cytokine response, *Pexa-Vec* stimulated differential chemokine secretion, including significantly higher secretion in comparison to baseline of C-X-C motif chemokine ligand (CXCL) -10, CXCL9, C-C motif ligand (CCL) -4, CCL7, and IL-16 in the Responder group at D2 (Figure 2A, Appendix A), albeit CCL4 remained at lower absolute levels in comparison to Exhausted patients. These chemokines are potent inducers of immune cell migration and tissue infiltration, including CD8 T cells [41], CD4 T cells [42], NK cells [43], and monocytes [44]. Chemokine secretion coincided with a transient lymphopenia in the peripheral blood of all patients at D2 after *Pexa-Vec* but was significantly more marked in Responder than Exhausted patients (Figure 2B). Peripheral blood lymphopenia is commonly observed following both therapeutic and pathogenic virus infection and is associated with the migration of lymphocytes to lymph nodes and to sites of tissue infection [45]. We examined changes in the proportions of lymphocyte subsets (CD4 and CD8 T cells, NK cells and NKT cells) between pre-*Pexa-Vec* and D2 levels and found a greater reduction across all subsets in the Responder group when compared to the Exhausted group, though this was only statistically significant in CD4 helper T-cells (Figure 2C). Of these immune subsets, CD8 T cells play a critical role in mediating OV immunotherapy [7,44,45]. CD8 T cells were detected in the resected tumour specimens from all available patient tumours following *Pexa-Vec* infusion. However, it is unclear as to whether the tissue-resident CD8 T cells were altered by *Pexa-Vec* due to lack of available pre-treatment biopsies. Taken together, these results indicate that Responder patients secrete higher concentrations of type IIFNs, inflammatory cytokines and chemokines following *Pexa-Vec* infusion, which is associated with peripheral blood lymphopaenia.

### 3.4. Differential Immune Cell Activation following Pexa-Vec Infusion

Based on the observed differences in inflammatory cytokines, we investigated the activation of peripheral immune cell subsets by way of cell surface expression of the early activation marker, CD69, which is upregulated in response to many cytokines, including IL-12, IL-18, type I IFN, and IL-2 [46,47,48,49]. As would be expected, Responder patients had greater elevation in CD69 immune cell expression than Exhausted patients at D2 (Figure 3A, Appendix A). CD69 expression was significantly different between the two groups on NK and NKT cells, which are known to mediate *Pexa-Vec* therapy [50,51]. We therefore tested the functional cytolytic capacity of patient-derived peripheral blood NK cells against tumour-relevant cell lines, revealing a significantly higher increase from baseline at D2 for the Responder group in comparison to the Exhausted group (Figure 3B). A further consequence of immune cell activation is the subsequent expression of immune checkpoint proteins, thereby temporarily limiting potentially harmful autoimmune effects [52]. Accordingly, fold-change in expression at D2 compared to baseline, of the cell surface immune checkpoint ligand, PD-L1, increased across the majority of immune cell subsets in the Responder group when compared to the Exhausted group (Figure 3C, Appendix A).

### 3.5. Baseline Cytokine Concentrations and Regulatory T Cells Predict the Immune Response to Pexa-Vec

We sought to identify baseline soluble factors predictive of the immunological response to *Pexa-Vec* infusion. Luminex quantification revealed a large number of inflammatory cytokines that were significantly higher at baseline in the Exhausted group, in comparison to the Responder group (Figure 4A). These included IL-1β, IL-9, IL-18, macrophage colony-stimulating factor (M-CSF), macrophage migration inhibitory factor (MIF), granulocyte colony-stimulating factor (G-CSF), and TNF-α, cytokines involved in inflammation and activation of immune cells [53,54,55], mobilisation, activation, and survival of myeloid cells [56] and DCs [57] and the promotion of anti-tumour immune responses [58,59,60]. In contrast, no inflammatory cytokines were significantly higher at baseline in the Responder group. Likewise, baseline chemokine concentrations were significantly higher in the Exhausted group, in comparison to the Responder group, as exemplified by IL-18, CXCL10, CXCL1, CCL2, and CCL4, which function as chemo-attractants to T cells [61,62,63,64,65] (Figure 4B). Both IL-18 and CCL4 were expressed at high levels at baseline and remained equally high post-treatment. In contrast, the Responder patients expressed these solutes at much lower levels at baseline, which increased following treatment, although post-treatment levels remained below the baseline levels observed in the Exhausted group. The exception to this trend was IL-16, which was significantly higher at baseline in the Responder group. IL-16 is also a chemoattractant specifically for helper CD4+ T cells [66]. 

Importantly, there was no overlap in the baseline concentrations of these cytokines between Responder and Exhausted patients, meaning that they can potentially each be utilised as highly sensitive and specific predictive markers for the immunological response to *Pexa-Vec i.v*. infusion. In accordance with the baseline secretion of inflammatory cytokines, PD-L1 expression was higher across the majority of PBMC subsets in the Exhausted group, in comparison to the Responder group, particularly in T cell subsets (Figure 4C). We also found that the baseline level of Treg cells, as a proportion of PBMCs, was significantly higher in the Exhausted group (Figure 4D).

## 4. Discussion

In this study, we have shown a distinct IFN-mediated peripheral blood immune response to *Pexa-Vec*; out of a total of nine patients treated within this study, four exhibited a significant and classical IFN-driven response, as would be anticipated following OV therapy. However, the remaining five patients appeared to exhibit a significantly lower amplitude of immune response and displayed a typical exhausted immune profile at baseline, with higher baseline Treg levels. Although post-treatment IFN responses to OVs are important in boosting the development of anti-cancer immunity, IFN signaling prior to treatment has been associated with reduced overall survival [22]. Higher pre-therapy IFN signaling represents chronic immune activation, with consequent immune exhaustion/suppression. Additionally, higher levels of baseline IFN signaling have been correlated with an anti-viral state in some cancers, thereby blocking the therapeutic efficacy of OVs [67,68].

Amongst the many obstacles linked with the development of new anticancer therapeutics is the absence of biomarkers predictive of a successful anti-cancer immune response. For OV therapies, soluble peripheral blood biomarkers predictive of clinical benefit that can easily be measured by non-invasive approaches, would help to significantly move the field forward. Many retrospective studies of immunotherapeutic OV trials have been performed in an attempt to identify biomarkers of response to therapy. Although some associations with clinical benefit and survival with OV have been identified, none have yet been validated as predictive biomarkers. One study showed that a greater baseline prevalence of circulating lymphocytes with greater proliferative capacity and pre-existing IFN signaling are associated with a lack of response to adenovirus [22]. Similar to our current findings, these features indicate that a pre-existing activated immune response and the resulting immune refractoriness may prevent a patient’s response to OV therapy.

Our results identify a panel of seven peripheral blood cytokines, two of which are TH1 cytokines (MIF and TNF-a), that can each be employed as biomarkers to predict subsequent immune cell activation, chemokine secretion, and NK cell cytolytic function upon i.v. vaccinia virus therapy. Many of these cytokines are IFN-stimulated genes, associated with NK and TH1 T cell activity, proliferation and chemokinesis. Baseline secretion levels of these cytokines were all higher in Exhausted patients, with no overlap in comparison to the much lower levels associated with Responder patients, indicating their utility as specific biomarkers to predict patient response to *Pexa-Vec* therapy.

As well as the divergent cytokine and cellular activation response between the two patient groups, we also present data signifying a differential chemokine pattern. CXCL10, CXCL9, CCL4, CCL7 and IL-16 are all involved in cellular migration from the periphery into tissues [39,40,41,42] and peaked at significantly higher levels in Responder patients at D2 following *Pexa-Vec* infusion. This D2 peak in chemokines was again absent in Exhausted patients, where baseline secretion of almost every chemokine was higher. Chemokine secretion in Responder patients only was associated with transient lymphopenia, in keeping with lymphopenia following pathogenic viral infection, which is associated with the migration of immune cells into lymph nodes and tissue sites of inflammation [45]. In addition to chemotaxis, CXCL10 has also been shown to modulate the activation of effector cells in both sites of inflammation and draining lymph nodes [22], specifically an involvement in T cell priming [42]. Intriguingly, baseline levels of IL-16 were higher in Responder patients, revealing a complex picture that requires confirmation in larger clinical trials. Unlike other cytokines and chemokines, IL-16 messenger RNA is constitutively expressed in T cells, eosinophils, and DCs. IL-16 pro-protein accumulates in these cells, ready for secretion upon stimulation. Higher baseline levels in Responder patients may therefore reflect a greater concentration of these cell types [69,70,71].

Mirroring expression of CD69, expression levels of PD-L1, increased to higher levels in the majority of Responder than Exhausted patient PBMC subsets following treatment. Whilst PD-L1 is immunosuppressive, its expression following therapy correlates with an anti-tumour response driven by IFNs and other inflammatory cytokines [72]. In fact, PD-L1 expression on both peripheral and tumour-infiltrating T cells in response to therapy is associated with better prognosis [73,74]. In contrast, pre-existing PD-L1 expression on immune cells in our trial was predictive of a chronically exhausted phenotype and the inability to respond to virus stimulus.

We discovered an elevated frequency of circulating Tregs in Exhausted patients, reinforcing the concept of immune exhaustion at baseline in these patients, which may prevent an effective immune response to immunotherapy. Depletion of Tregs prior to *Pexa-Vec*, e.g., via anti-CTLA-4/anti-CD25 therapy, may enable a more robust response to subsequent *Pexa-Vec*/PD-L1 therapy, paving the way for personalised treatment strategies. Alternatively, oncolytic virotherapy administered at much earlier stages of planned cancer treatment, before the immune system is exhausted, may be more effective than if given at later stages when the TME is more suppressed.

## 5. Conclusions

In summary, we have shown that OV therapy induces a classical IFN response in Responder patients, including the release of inflammatory cytokines/chemokines, which can activate T cells, as evident from elevated CD69 levels. These activated T cells can then infiltrate into tumours, with the potential to kill tumour cells. In contrast, higher levels of cytokines and chemokines were present at baseline in the Exhausted group, alongside elevated CD69 and PD-L1, which were not upregulated further post-treatment, with some being maintained at high levels. Although our patient cohort is limited in size, the data shown are well-defined, with largely non-overlapping differential responses between the two groups. Verification of these potential biomarkers in both tumours and peripheral blood require further consideration in larger clinical trials in order to correlate TME and clinical outcomes.

## Figures and Tables

**Figure 1 cancers-14-02181-f001:**
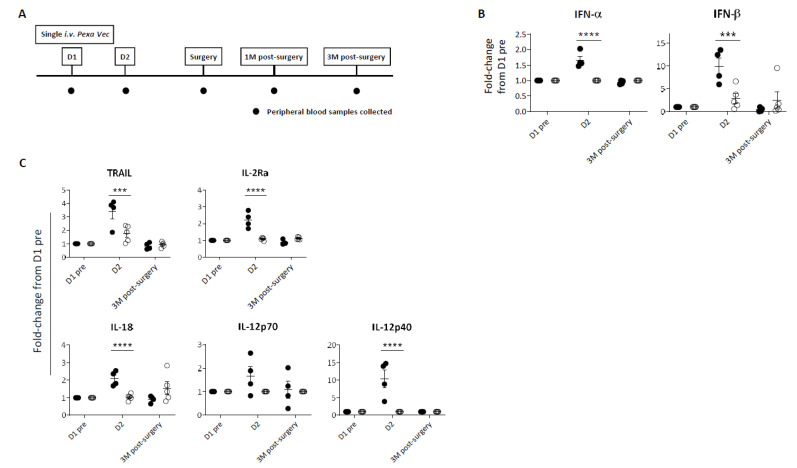
Peripheral immune response to *Pexa-Vec* infusion. (**A**) Trial schema showing timing of virus infusion and peripheral blood sample collection. Differential IFN (**B**) and inflammatory cytokine (**C**) response to *Pexa-Vec* in Responder (black; *n* = 4) and Exhausted (white; *n* = 5) patients. Data is shown as fold-change from baseline (D1 pre); *** *q*-value < 0.001, **** *q*-value < 0.0001.

**Figure 2 cancers-14-02181-f002:**
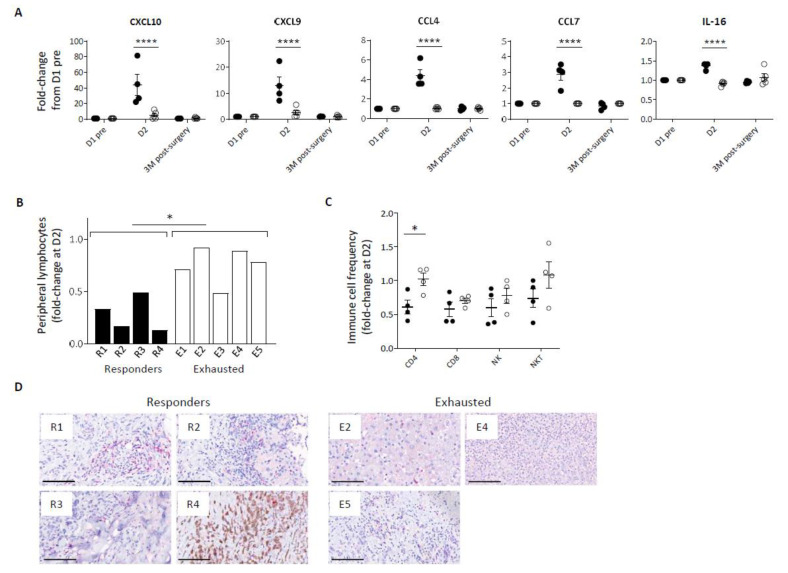
Immune cell redistribution in response to *Pexa-Vec* infusion. (**A**) Differential chemokine response in Responder (black; *n* = 4) and Exhausted (white; *n* = 4/5) patient plasma to *Pexa-Vec* infusion. Data is shown as fold-change difference from baseline (D1 pre) (**** *q*-value < 0.0001). (**B**) Total lymphocyte count and (**C**) individual immune cell populations (CD4^+^ T cells, CD8^+^ T cells, NK cells, NKT cells) both represented by fold-change difference between D2 and baseline (D1 pre) in Responder (black; *n* = 4) and Exhausted (white; *n* = 5 (**A**) and *n* = 4 (**B**)) patients (* *p* < 0.05). (**D**) Representative images of CD8 T cells in Responder and Exhausted tumour (CD8-positive cells are visualised by Fast Red staining). Bars represent 100 µm.

**Figure 3 cancers-14-02181-f003:**
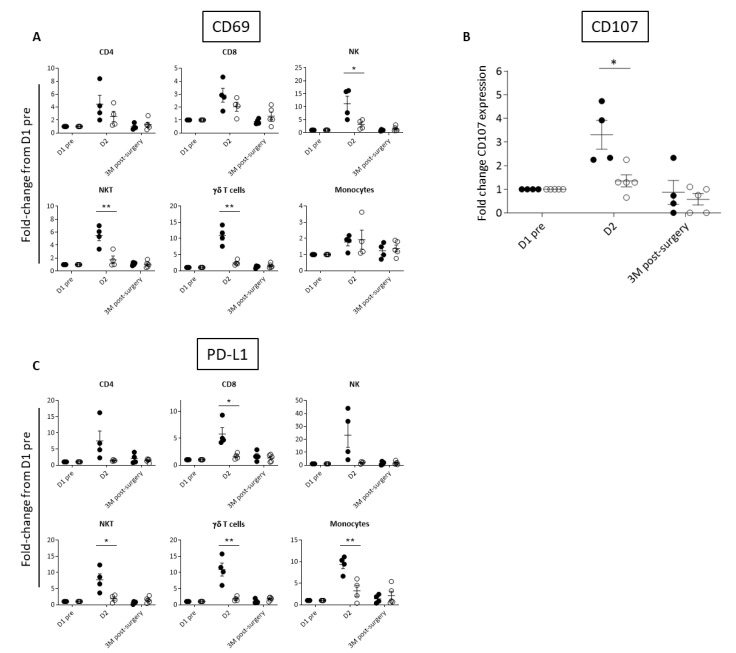
Differential activation of immune cell populations following *Pexa-Vec* infusion. Differential expression of (**A**) CD69 in immune cell populations, (**B**) NK CD107 expression (representing NK cell degranulation), and (**C**) PD-L1 expression in immune cell populations in Responder (black; *n* = 4) and Exhausted (white; *n* = 4/5) patients at baseline (D1 pre) and following *Pexa-Vec* infusion. Data is shown as fold-change difference from baseline (D1 pre) for % positive expression (* *p* < 0.05, ** *p* < 0.01).

**Figure 4 cancers-14-02181-f004:**
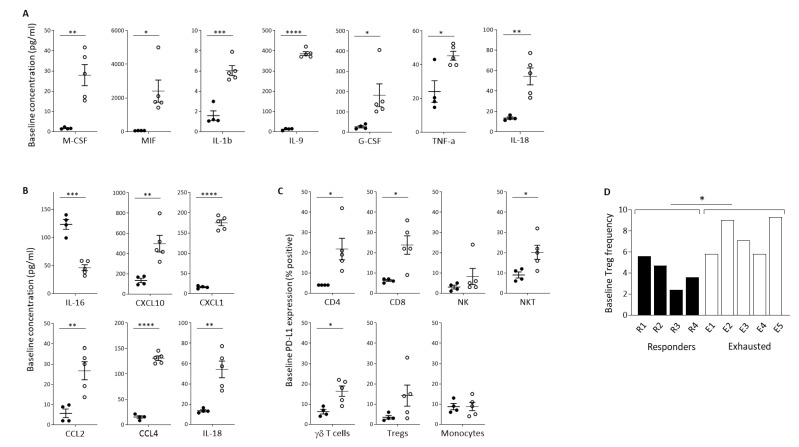
Baseline indicators of predicted response to *Pexa-Vec* therapy. Differential baseline levels of (**A**) inflammatory cytokines and (**B**) chemokines in patient plasma and (**C**) PD-L1 expression on immune cell populations in Responder (black; *n* = 4) and Exhausted (white; *n* = 5) patient samples. Data is shown as pg/mL (**A**,**B**) or % positive expression for D1 pre samples (* *p* < 0.05, ** *p* < 0.01, *** *p* < 0.001, **** *p* < 0.0001). (**D**) Relative frequency of Tregs in PBMCs at baseline (D1 pre) in Responder (black; *n* = 4) and Exhausted (white; *n* = 5) patients. Data is shown as % Tregs of whole PBMCs; * *p* < 0.05.

**Table 1 cancers-14-02181-t001:** Patient demographics. Patient details are listed, including Responder (R) or Exhausted (E) group, age, gender, cancer type, and the sum of the longest diameters of the tumours for each patient.

Patient	Age	Gender	Cancer	Sum of Longest Diameters of Tumours (mm)
R1	71	M	Melanoma	76
R2	63	M	CRCLM	18
R3	74	F	Melanoma	10
R4	64	F	Melanoma	60
E1	65	F	CRCLM	10
E2	74	F	CRCLM	27
E3	47	F	CRCLM	N/A *
E4	59	M	CRCLM	25
E5	79	F	CRCLM	80

* E3 did not have planned surgery due to disease progression.

## Data Availability

The data presented in this study are available on request from the corresponding author.

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
