# Peer review of "Intravenous Oncolytic Vaccinia Virus Therapy Results in a Differential Immune Response between Cancer Patients"

_cancers, 2022, doi:10.3390/cancers14092181_

Round 1
Reviewer 1 Report
Interesting article overall. Nice work to lay the foundation for further studies in identifying patients who may or may not benefit from oncolytic virotherapy.
- Have you consider genetic testing as well to see if that may reliably predict which patients may respond?
- Do you think this data could be applicable to other oncolytic viruses or just pexa-vec?
- Does this make an argument to give oncolytic virotherapy earlier in cancer treatment courses before the immune system is "exhausted"?
- Would be interesting to examine immune response along with circulating tumor DNA and agree we need to examine this along with clinical response.
Author Response
REVIEWER 1 - Comments and Suggestions for Authors
Interesting article overall. Nice work to lay the foundation for further studies in identifying patients who may or may not benefit from oncolytic virotherapy.
We thank the reviewer.
- Have you consider genetic testing as well to see if that may reliably predict which patients may respond?
We agree that genetic testing could pave the way for predicting response to oncolytic virus therapy. We have included a sentence in the introduction, indicating that OV trials are currently recruiting patients with dMMR tumours, as it is thought that a higher tumour-mutation burden would lead to higher tumour immunogenicity and better anti-cancer effects. Future translational studies will address this concept.
1.2 Do you think this data could be applicable to other oncolytic viruses or just pexa-vec?
As mentioned in the discussion section, further trials are required to determine if the same proteins can be used as biomarkers to predict immune responses. We are actively pursuing this strategy.
- Does this make an argument to give oncolytic virotherapy earlier in cancer treatment courses before the immune system is "exhausted"?
We agree that giving OV therapy earlier in cancer treatment may be more beneficial and have added a sentence in the discussion.
- Would be interesting to examine immune response along with circulating tumor DNA and agree we need to examine this along with clinical response.
Thank you, we agree with the reviewer.
Reviewer 2 Report
In this manuscript, the authors treated total 9 cancer patients with oncolytic virus Pexa-Vec and analyzed peripheral blood levels of inflammatory cytokines and chemokines, immune cell counts, and expression of CD69 and PD-L1 on the cell surface of immune cells. They demonstrated that many cytokines and chemokines were upregulated on Day2 in fold-change in Responder group than Exhausted group, CD69 and PD-L1 expression in immune cells was also upregulated in fold-change, but the baseline level of most of the cytokines and chemokines as well as PD-L1 expression in most immune cells were higher in Exhausted group than Responder group. On Day2, CD4 count significantly decreased in Responder group, and Treg count at the baseline was higher in Exhausted group. Coculture assay of tumor cells and PBMCs showed an increase of CD107 expression on Day2 in Responder group.
The authors want to say that there is a differential response in the Responder and Exhausted group and high baseline blood cytokine and chemokine levels and high PD-L1 levels in T cells might be predictive biomarkers for the Pexa-Vec (negative) response, however, in addition to the small size of this study, the interpretation might be somewhat misleading.
1) Firstly, about the definition of Responder group and Exhausted group. Among the 9 samples, 4 showed the classical response of interferon and 5 did not. So, they defined the former as “Responder” group and the latter as “Exhausted” group. The authors are describing it in the last part of Introduction, but it should be included in the Results section, along with Figure 1.
2) Patients characteristics, age, gender, performance status (PS), etc., and the prognosis, should be shown. Was the prognosis of Responder group better than Exhausted group?
3) In Figures 1 – 3, data were shown as fold change, but in Figure 4 absolute value was shown. In Figures 1-3, the authors described that Responder group showed significantly higher fold change of certain inflammatory cytokines and chemokines in the peripheral blood, as well as that of CD69 and PD-L1 expression in certain immune cells on Day2, but baseline absolute value of the cytokines and chemokines in the peripheral blood as well as PD-L1 expression in immune cells was significantly higher in Exhausted group. Therefore, for example, CCL4 increased three-fold in Response group in Figure 2A, but it is much lower in the absolute value than Exhausted group according to Figure 4B. IL-18 increased two-fold in Responder group in Figure 1C, but it is much lower in the absolute value than Exhausted group in Figure 4B. PD-L1 expression on the CD8+ T cells increased six-fold in Responder group in Figure 3C, but it might not be so different in the absolute value compared with Exhausted group according to Figure 4C. Considering these, the authors should re-analyze by using the absolute value, not by fold change, otherwise it is misleading.
4) The re-analysis mentioned above might change the whole view of this study. In addition, the authors sometimes described, “PD-L1 increased across all immune cell subsets in the Responder group” in Line 296, and “PD-L1 expression was higher across all PBMC subsets in the Exhausted group” in Line 324, however, statistical significance was not shown in all the subsets. Scientifically, no statistical significance means “no increase” and “not higher”. The authors should review the whole manuscript and use scientifically correct description.
5) In the Introduction and Abstract, the authors described that OVs’ effect is to change the “cold” tumor to “hot” tumor. Considering this, “a pre-existing activated immune response” described in the Discussion in Line 359 seems to be “hot” tumor microenvironment and it seems inconsistent as the finding in Exhausted group. Can the authors give some explanation?
6) PD-L1 is usually expressed on the tumor cells and binds with PD-1 on the immune cells. What is the function of PD-L1 on the immune cells? Two references were cited, but Ref. 73 seems describing PD-L1 on the tumor cells.
7) Introduction is too long relative to the following text and had better be condensed.
8) What markers were used to define NK, NKT and monocytes in this study? The authors should clearly describe in the Materials and Methods.
9) Is there reliable references of NK cell degranulation assay? Figure 3B might be the degranulation assay, but the figure legend seems not explaining it.
10) Finally, this is a very small sized study, which is a fundamental limitation. The authors had better increase the size and re-analyze including the prognosis.
Author Response
REVIEWER 2 - Comments and Suggestions for Authors
In this manuscript, the authors treated total 9 cancer patients with oncolytic virus Pexa-Vec and analyzed peripheral blood levels of inflammatory cytokines and chemokines, immune cell counts, and expression of CD69 and PD-L1 on the cell surface of immune cells. They demonstrated that many cytokines and chemokines were upregulated on Day2 in fold-change in Responder group than Exhausted group, CD69 and PD-L1 expression in immune cells was also upregulated in fold-change, but the baseline level of most of the cytokines and chemokines as well as PD-L1 expression in most immune cells were higher in Exhausted group than Responder group. On Day2, CD4 count significantly decreased in Responder group, and Treg count at the baseline was higher in Exhausted group. Coculture assay of tumor cells and PBMCs showed an increase of CD107 expression on Day2 in Responder group.
The authors want to say that there is a differential response in the Responder and Exhausted group and high baseline blood cytokine and chemokine levels and high PD-L1 levels in T cells might be predictive biomarkers for the Pexa-Vec (negative) response, however, in addition to the small size of this study, the interpretation might be somewhat misleading.
We thank the reviewer for their comments. With regard to the sample size, we agree that this is small, as mentioned in the introduction and discussion sections. We agree that confirmation in larger studies is warranted if these findings are to be applied as predictive biomarkers. We also feel that there is a distinct difference in the Responder and Exhausted groups, with many statistically significant findings.
- Firstly, about the definition of Responder group and Exhausted group. Among the 9 samples, 4 showed the classical response of interferon and 5 did not. So, they defined the former as “Responder” group and the latter as “Exhausted” group. The authors are describing it in the last part of Introduction, but it should be included in the Results section, along with Figure 1.
We have now included this in the Results section in association with Figure 1.
2.2 Patients characteristics, age, gender, performance status (PS), etc., and the prognosis, should be shown. Was the prognosis of Responder group better than Exhausted group?
We have now included a table (Table 1) showing patient characteristics, age, gender and the sum of the longest tumour diameters, for each patient, which can be used as a guide to prognosis; the tumour volumes are similar across the two patient groups. We have also mentioned the cited prognosis of surgically resected CRCLM and metastatic melanoma in the introduction.
2.3 In Figures 1 – 3, data were shown as fold change, but in Figure 4 absolute value was shown. In Figures 1-3, the authors described that Responder group showed significantly higher fold change of certain inflammatory cytokines and chemokines in the peripheral blood, as well as that of CD69 and PD-L1 expression in certain immune cells on Day2, but baseline absolute value of the cytokines and chemokines in the peripheral blood as well as PD-L1 expression in immune cells was significantly higher in Exhausted group. Therefore, for example, CCL4 increased three-fold in Response group in Figure 2A, but it is much lower in the absolute value than Exhausted group according to Figure 4B. IL-18 increased two-fold in Responder group in Figure 1C, but it is much lower in the absolute value than Exhausted group in Figure 4B. PD-L1 expression on the CD8+ T cells increased six-fold in Responder group in Figure 3C, but it might not be so different in the absolute value compared with Exhausted group according to Figure 4C. Considering these, the authors should re-analyze by using the absolute value, not by fold change, otherwise it is misleading.
As described in the manuscript, the Responder group are characterised by a non-activated immune profile at baseline, which significantly changes with treatment. In contrast, the Exhausted group are characterised by an activated and exhausted immune profile at baseline, which changes relatively little with treatment. The fold change from baseline is therefore used to show this difference between the groups, as it is the change from baseline that is of importance. This fold-change is a common strategy used previously with similar data set, Ribas et al, Cell. 2017;170(6), which we have clarified in section 2.10. We have now provided Supplementary Figures (Suppl. Fig. 1 & 2) showing the absolute values for the cytokine and immunophenotyping data that is shown as fold-change in the Results section. Annotations to the Results section now also includes reference to these Supplementary Figures. The absolute values largely support the fold-change values
2.4 The re-analysis mentioned above might change the whole view of this study. In addition, the authors sometimes described, “PD-L1 increased across all immune cell subsets in the Responder group” in Line 296, and “PD-L1 expression was higher across all PBMC subsets in the Exhausted group” in Line 324, however, statistical significance was not shown in all the subsets. Scientifically, no statistical significance means “no increase” and “not higher”. The authors should review the whole manuscript and use scientifically correct description.
We thank the reviewer for this comment and have amended the manuscript accordingly.
2.5 In the Introduction and Abstract, the authors described that OVs’ effect is to change the “cold” tumor to “hot” tumor. Considering this, “a pre-existing activated immune response” described in the Discussion in Line 359 seems to be “hot” tumor microenvironment and it seems inconsistent as the finding in Exhausted group. Can the authors give some explanation?
Indeed, we frequently find that OV therapy does change a cold tumour to hot, but that this is associated with exhaustion, as exemplified by increased PD-L1 expression. We have therefore suggested that the patients with an Exhausted immune profile at baseline might be treated with an immune checkpoint inhibitor at baseline, to partially reverse this exhaustion.
2.6 PD-L1 is usually expressed on the tumor cells and binds with PD-1 on the immune cells. What is the function of PD-L1 on the immune cells? Two references were cited, but Ref. 73 seems describing PD-L1 on the tumor cells.
Ref. 73 indicates that there is a prognostic significance of PD-L1 expression on both tumour and immune cells within the TME. Moreover, expression of PD-L1 significantly correlates with increasing densities of immune cells within the tumour. This ties in with the response to reviewer comment 2.5 whereby we suggest that Exhausted patients may benefit from a pre-treatment of PD-L1 inhibitor prior to OV therapy.
2.7 Introduction is too long relative to the following text and had better be condensed.
We have shortened this (see attached manuscript).
2.8 What markers were used to define NK, NKT and monocytes in this study? The authors should clearly describe in the Materials and Methods.
The markers used to define the cell populations have been included in the relevant Materials & Methods sections.
2.9 Is there reliable references of NK cell degranulation assay? Figure 3B might be the degranulation assay, but the figure legend seems not explaining it.
The title in the Materials & Methods section has been changed to include ‘CD107’ to clarify and link to the relevant results section. A citation has also been included as a reference for this established assay.
2.10 Finally, this is a very small sized study, which is a fundamental limitation. The authors had better increase the size and re-analyze including the prognosis.
We agree that this was a small study and it is now closed precluding further patient recruitment. Further confirmation of the findings is warranted in larger patient cohorts in future trials, as discussed in the manuscript. The results described here appear to show a distinct difference between Responder and Exhausted groups, despite the small sample size.
Reviewer 3 Report
Dear editor,
In the manuscript entitled “Intravenous oncolytic vaccinia virus therapy results in a differential immune response between cancer patients” the authors analyzed patient response to i.v oncolytic virotherapy, with emphasis on differential immune response to identify biomarkers for therapy efficiency. The paper presents the central role of baseline indicators involved in the concept of immune exhaustion, and points its relevance towards personalized treatment.
General aspects:
- The results are new, interesting, and relevant. The paper is well written, and the results are clearly presented.
- The manuscript meets the quality standard of the journal. Thus, this reviewer RECOMMENDS the publication in Cancers.
Author Response
|
REVIEWER 3 - Comments and Suggestions for Authors
|
|
|
|
|
|
Dear editor,
In the manuscript entitled “Intravenous oncolytic vaccinia virus therapy results in a differential immune response between cancer patients” the authors analyzed patient response to i.v oncolytic virotherapy, with emphasis on differential immune response to identify biomarkers for therapy efficiency. The paper presents the central role of baseline indicators involved in the concept of immune exhaustion, and points its relevance towards personalized treatment.
General aspects:
- The results are new, interesting, and relevant. The paper is well written, and the results are clearly presented.
- The manuscript meets the quality standard of the journal. Thus, this reviewer RECOMMENDS the publication in Cancers.
We thank the reviewer for their comments.
Round 2
Reviewer 2 Report
Comments to the authors responses
2.1 This study observed that 4 patients showed the classical response of interferon and 5 did not, and defined the former as “Responder” group and the latter as “Exhausted” group. In this version, the authors included this in the Results section, but it is also still included in the Introduction. The redundant description had better be removed.
2.2 The authors added patients’ characteristics in Table 1, but there is no description about tumor type (melanoma or CRC), which should be noted. Did any patients receive previous therapy? It should be clearly described (maybe in the Materials and Methods), because it might surely affect the tumor immune microenvironment. This study seems registered in 2012, according to the EudraCT number, so, probably prognosis of the patients (overall survival or recurrent-free survival after the surgery) can be included and the association between the response to the OV and the prognosis might be important.
2.3 - 2.5 The authors added absolute value data of cytokines and CD69&PD-L1 expression in Supplementary Figures 1 and 2. According to the data, upregulation and absolute value of CD69 and PD-L1 expression at Day2 in the Responder group seems significant compared to the Exhausted group in NK, NKT, γδT as well as PD-L1 in monocytes in Supplementary Figure 2. Statistical significance of the absolute value between the Responder and Exhausted at Day2 might help for readers to understand. The authors said that fold change is important, however, it is difficult to simply accept according to the abundance of IL-18 and CCL4 in Supplementary Figure 1. The authors should be more careful to interpret these data. At least, IL-18 and CCL4 should not be handled in the same way with other cytokines examined in this study.
Initial hypothesis might be: OV injection induces classical interferon response including ILs, then immune cells are activated with upregulated CD69 expression and infiltrate into the tumors, thereby possibly killing the tumor cells, in the Responder group. In addition, this study also observed that the Exhausted group showed significantly higher cytokines and chemokines at the baseline and showed infrequent upregulation of the cytokines and chemokines as well as CD69 and PD-L1 in fold change, although some of them were maintained at high level. Important thing is how these responses to OV affect the prognosis after the surgery, however, it is lacking in this study. If the authors can add the prognosis data in Table 1, this study can have more discussion.
2.9 Figure 3B is a different assay from Figure 3A and C. The legend should clearly explain it.
Author Response
- This study observed that 4 patients showed the classical response of interferon and 5 did not, and defined the former as “Responder” group and the latter as “Exhausted” group. In this version, the authors included this in the Results section, but it is also still included in the Introduction. The redundant description had better be removed.
We thank the reviewer for their further comments and insightful suggestions.
We have removed this description from the introduction.
- The authors added patients’ characteristics in Table 1, but there is no description about tumor type (melanoma or CRC), which should be noted. Did any patients receive previous therapy? It should be clearly described (maybe in the Materials and Methods), because it might surely affect the tumor immune microenvironment. This study seems registered in 2012, according to the EudraCT number, so, probably prognosis of the patients (overall survival or recurrent-free survival after the surgery) can be included and the association between the response to the OV and the prognosis might be important.
We have now included the tumour type in Table 1. With regard to the prognostic follow up of patients, the primary endpoint of this trial was the safety and feasibility of a single infusion of an oncolytic virus ahead of planned surgical resection. Survival outcomes were therefore not considered within this study, in contrast to therapeutic OV clinical trials with an efficacy endpoint, where patients receive multiple doses of virus.
Multiple factors determine whether cancer recurrence occurs following surgical resection, one of the most important of which is the size of the resected tumours, which we have included in the table.
- The authors added absolute value data of cytokines and CD69&PD-L1 expression in Supplementary Figures 1 and 2. According to the data, upregulation and absolute value of CD69 and PD-L1 expression at Day2 in the Responder group seems significant compared to the Exhausted group in NK, NKT, γδT as well as PD-L1 in monocytes in Supplementary Figure 2. Statistical significance of the absolute value between the Responder and Exhausted at Day2 might help for readers to understand. The authors said that fold change is important, however, it is difficult to simply accept according to the abundance of IL-18 and CCL4 in Supplementary Figure 1. The authors should be more careful to interpret these data. At least, IL-18 and CCL4 should not be handled in the same way with other cytokines examined in this study.
We have now included the statistical significance of the data in Supplementary Figure 2 for the comparison of the absolute values between the Responder and Exhausted patients at day 2. With regard to the differences between the majority of the cytokines and IL-18/CCL4 in Exhausted patients shown in Supplementary Figure 1, we have included the following sentence in the manuscript: “both IL-18 and CCL4 were expressed at high levels at baseline and remained equally high post-treatment. In contrast, the Responder patients expressed these solutes at much lower levels at baseline, which increased following treatment, although post-treatment levels remained below the baseline levels observed in the Exhausted group.”
- Initial hypothesis might be: OV injection induces classical interferon response including ILs, then immune cells are activated with upregulated CD69 expression and infiltrate into the tumors, thereby possibly killing the tumor cells, in the Responder group. In addition, this study also observed that the Exhausted group showed significantly higher cytokines and chemokines at the baseline and showed infrequent upregulation of the cytokines and chemokines as well as CD69 and PD-L1 in fold change, although some of them were maintained at high level. Important thing is how these responses to OV affect the prognosis after the surgery, however, it is lacking in this study. If the authors can add the prognosis data in Table 1, this study can have more discussion.
We thank the reviewer for their suggested concluding remarks, which we have incorporated into the manuscript. Unfortunately, we are unable to provide any additional prognostic data for the patients; please see comments to #2 above.
2.9 Figure 3B is a different assay from Figure 3A and C. The legend should clearly explain it.
Figure 3 legend has now been modified to reflect the different assays shown in the figure.
Round 3
Reviewer 2 Report
The authors adequately responded to most of the comments .
Still I'm not sure how much IL-18 and CCL4 upregulation contributes in the Responder group. According to Fig. 1&2 and Suppl. Fig. 1, the authors may note the upregulation of IFNα, IL-12p40, CXCL10, CXCL9, CCL7 and IL-16, but had better omit the description about the upregulation of IFNβ, TRAIL, IL-2Rα, IL-18 and CCL4, since the absolute value is similar or lower in the Responder group.
Author Response
The authors adequately responded to most of the comments .
Still I'm not sure how much IL-18 and CCL4 upregulation contributes in the Responder group. According to Fig. 1&2 and Suppl. Fig. 1, the authors may note the upregulation of IFNα, IL-12p40, CXCL10, CXCL9, CCL7 and IL-16, but had better omit the description about the upregulation of IFNβ, TRAIL, IL-2Rα, IL-18 and CCL4, since the absolute value is similar or lower in the Responder group.
We thank the reviewer for their comments. We have included several modifications to the wording in the results section, describing that although a peak post-treatment is observed in IFNβ, TRAIL, IL-2Rα, IL-18 and CCL4, indicating an inflammatory response in Responder patients, the absolute values remain similar to, or below those observed in the Exhausted patients.